# The Presence of Hypoechoic Micronodules in Patients with Hashimoto′s Thyroiditis Increases the Risk of an Alarming Cytological Outcome

**DOI:** 10.3390/jcm10040638

**Published:** 2021-02-07

**Authors:** Dorota Słowińska-Klencka, Martyna Wojtaszek-Nowicka, Mariusz Klencki, Kamila Wysocka-Konieczna, Bożena Popowicz

**Affiliations:** 1Department of Morphometry of Endocrine Glands, Chair of Endocrinology, Medical University of Lodz, Pomorska Str 251, 92-213 Łódź, Poland; marklen@tyreo.umed.lodz.pl (M.K.); kamila.wysocka91@wp.pl (K.W.-K.); bozena.popowicz@umed.lodz.pl (B.P.); 2Department of Clinical Endocrinology, Medical University of Lodz, Pomorska Str 251, 92-213 Łódź, Poland; martyna.wojtaszek-nowicka@umed.lodz.pl

**Keywords:** thyroid, Hashimoto′s thyroiditis, thyroid cancer, ultrasonography

## Abstract

The aim of the study was to identify a possible relation between various ultrasonographic (US) appearances of Hashimoto′s thyroiditis (HT) and the risk of obtaining an alarming cytology of coexisting nodules. The study included 557 patients with HT, who had been referred for fine needle aspiration biopsy (FNA). We divided US patterns of HT (UP-HT) into eight groups: (a) Hypoechoic (compared to submandibular glands), homogeneous/fine echotexture; (b) hypoechoic, heterogeneous/coarse echotexture; (c) marked hypoechoic (darker than strap muscles), heterogeneous/coarse echotexture; (d) heterogeneous echotexture with hyperechoic, fibrous septa; (e) multiple, discrete marked hypoechoic areas (sized as 1 to 6 mm); (f) normoechoic pseudo-nodular areas; (g) echostructure similar to connective tissue; (h) thyroid parenchyma with no signs of HT. Indications for a surgical treatment resulting from the FNA outcome (categories IV–VI of Bethesda System for Reporting Thyroid Cytopathology) were identified only in patients with variants b, c, and e of UP-HT, but merely the “multiple, discrete marked hypoechoic areas” variant significantly increased the odds of obtaining such cytology (OR:5.7). The presence of the “normoechoic pseudo-nodular areas” variant significantly increased the odds for the benign cytology (OR:1.7). There are significant differences in the frequency of obtaining an alarming cytology in relation to the UP-HT variant.

## 1. Introduction

Hashimoto′s thyroiditis (HT), also called chronic lymphocytic or autoimmune thyroiditis, is the most common autoimmune endocrine disease, as well as the most common cause of hypothyroidism. HT is characterized by a progressive loss of thyroid follicular cells and lymphocytic infiltration of the thyroid parenchyma associated with fibrosis. It is believed that HT develops in genetically predisposed individuals who are exposed to an environmental trigger. The incidence of HT is higher in areas rich in iodine than in iodine-deficient regions. It is also higher among women than men [1,2]. The diagnosis of HT is based on clinical symptoms that result mainly from the development of hypothyroidism, serological tests (measurement of anti-thyroid antibodies), assessment of serum concentrations of thyroid stimulating hormone (TSH) and thyroid hormones, as well as ultrasound examination. A combination of the above-mentioned methods brings the best sensitivity and specificity for diagnosing HT. However, the clinical, serological, and sonographic presentation is highly variable as all these features of HT are present concurrently in less than 30% of cases [1,2,3]. An elevated level of serum anti-thyroid peroxidase antibodies (TPOab) is widely considered to be the best serological marker of HT. It is detectable in about 95% of patients with a clinically overt disease [2]. The specificity of TPOab in detecting HT verified with the microscopic examination (cytology or histopathology) approaches 90% [4].

The ultrasonographic (US) appearance of HT involves a wide range of findings. The thyroid may be hypoechoic, with a coarse, heterogeneous parenchymal echotexture, presence of the marginal abnormality, echogenic septations, multiple discrete hypoechoic micronodules or pseudo-nodular structures [5,6,7]. These features may be present separately or in different sets and can evolve over the course of the disease [5,6,7,8,9,10]. In some cases, discrete nodules may occur. A diverse and evolving ultrasound image of HT makes it difficult to differentiate between thyroid nodules and pseudonodules and hampers the optimal selection of lesions for fine needle aspiration biopsy (FNA) [11]. This is particularly disadvantageous in the context of higher prevalence of malignant and equivocal cytology (suspicious or indeterminate) in nodules accompanying HT when compared to nodules in patients without HT [12,13,14]. The relatively frequent occurrence of the alarming cytology is a consequence of higher incidence of papillary carcinoma (PTC) in patients with HT, as well as difficulties in the interpretation of microscopic images due to the common anisocytosis of thyroid follicular cells, their oxyphilic metaplasia, and variable inflammatory infiltration [12,13,14,15,16]. The above-mentioned difficulties in the morphological diagnosis inspired us to identify a possible relation between different US patterns of HT (UP-HT) and the risk of obtaining an alarming outcome from the biopsy of coexisting nodules that prompts the surgical treatment. According to our knowledge, no such analyses have been performed before. Usually, the published studies in that area have focused on the efficacy of the US examination in diagnosing HT [17,18,19]. The identification of UP-HT that differ in the risk of malignancy of co-existing nodules could lead to the optimization of diagnostic procedures in patients with this type of thyroiditis.

## 2. Material and Methods

### 2.1. Patients

FNA and US examinations were performed in a single center, in the years 2018–2020, in patients referred by endocrinologists from outpatient clinics. Patients with the diagnosis of HT confirmed by increased concentrations of TPOab, who had been referred for FNA of a co-existing nodule were enrolled in the study. Exclusion criteria were: (a) Positive history of Graves′ disease or increased concentration of antibodies against the TSH receptor, (b) surgical or radioiodine thyroid treatment in the past, (c) positive neck irradiation history. Overall, the study included 557 patients, including 27 (4.8%) males, mean age ± SD: 57.4 ± 14.1 years. Among them, 361 patients (64.8%) were treated with levothyroxine due to hypothyroidism and the others presented a normal thyroid function.

### 2.2. US and FNA Examinations

All the patients underwent a standard US examination of the thyroid with a detailed description of nodules subjected to FNA and an additional determination of the US pattern of thyroid parenchyma. We divided UP-HT into eight groups: (a) Hypoechoic (compared to submandibular glands), homogeneous/fine echotexture; (b) hypoechoic, heterogeneous/coarse echotexture; (c) marked hypoechoic (darker than strap muscles), heterogeneous/coarse echotexture; (d) heterogeneous echotexture with hyperechoic, fibrous septa; (e) multiple, discrete marked hypoechoic areas (micronodules—sized as 1 to 6 mm); (f) normoechoic pseudo-nodular areas; (g) echostructure similar to the connective tissue; h) thyroid parenchyma with no signs of HT (Appendix A). Additionally, we determined whether HT features were present in the whole thyroid (diffuse HT) or only a part of it (focal HT). When two or more patterns coexisted, the dominant pattern was assigned. The eighth variant of UP-HT (“thyroid parenchyma with no signs of HT”) was regarded as a type of diffuse HT. We also noted the presence of hypo- or hypervascularity of the gland in Power Doppler imaging, as well as lymph nodes typical of HT. On the basis of our experience and published data, we presupposed that such nodules show at least one of the following features: Round shape, hypoechoic image or absence of the hilum but without features highly suggestive of malignancy: Microcalcifications, cystic aspect, peripheral vascularity or diffusely increased vascularization or hyperechoic tissue looking as thyroid and short axis ≥8 mm [20,21,22,23]. Lymph nodes were evaluated at infrathyroidal and pretracheal areas and along the carotid arteries and jugular veins. The US examinations were performed by four experienced sonographers (with minimum 10 years′ experience), with the use of the Aloka Prosound Alpha 7 ultrasound system, ALOKA co. Ltd., Tokyo, Japan with a 7.5–14 MHz linear transducer. In the case of doubts, the ultrasound image was consulted by a single investigator (DSK). 

The biopsy was performed on thyroid nodules with a diameter of at least 5 mm (and usually over 1 cm) and at least one malignancy risk factor (US or clinical), according to the recommendations in effect in our country [24,25]. Nodules <1 cm were biopsied only if they showed particularly a worrying ultrasound image and none of their diameters were under 5 mm. Biopsies were performed following regular procedures. In most cases, two aspirations of a nodule were performed. Smears were fixed with a 95% ethanol solution and stained with haematoxylin and eosin. The FNA outcome of each nodule was classified into one of six categories in the Bethesda System for Reporting Thyroid Cytology (BSRTC). In this classification, category I includes non-diagnostic/unsatisfactory biopsies, category II: Benign lesions, category III: Follicular lesion of undetermined significance (FLUS) or atypia of undetermined significance (AUS), category IV: Suspicious for a follicular neoplasm, category V: Suspicious for malignancy, and category VI: Malignant neoplasm. Categories IV, V, and VI are regarded as an indication for surgical treatment due to the high risk of malignancy. Category III presents the most diverse risk of malignancy. At our center, similar to the majority of others, that risk is significantly higher for diagnoses of AUS than FLUS. A detailed description of the classification of nodules into specific diagnostic categories of the Bethesda system, as well as the risk of malignancy related to particular categories at our center were presented in our earlier report [26]. When one patient underwent FNA of two or more nodules, the nodule diagnosed with the highest BSRTC category was considered for the analysis of a relation between the FNA outcome and UP-HT. 

### 2.3. Analyses, Statistical Evaluation

Frequencies of identified UP-HT were evaluated in the examined group of patients. The mean age of patients, percentage of males, number of patients treated with levothyroxine, thyroid volume, average number of thyroid nodules revealed in a gland, average number of biopsied nodules in a patient, volume of those nodules, and the frequency of biopsied nodules with a diameter under 1 cm were determined for each variant of UP-HT. Additionally, the following features were compared between particular variants of UP-HT: Incidence of diffuse and focal presentation of HT, presence of lymph nodes typical of HT, hypervascularity and hypovascularity of thyroid parenchyma.

At the next step, the relation between the type of UP-HT and the category of FNA outcome was analyzed. The incidence of microscopic features of HT in smears with the diagnostic cellular material (presence of lymphocytic infiltration and oxyphilic metaplasia of follicular cells) was determined in relation to the UP-HT variant. Frequencies of particular BSRTC categories of cytological outcomes, especially those indicating the necessity of thyroid surgery, were compared between the examined variants of UP-HT. A cytological indication for the surgical treatment was assessed in two options: (1) When only categories IV-VI of BSRTC were regarded as an indication for the surgery, or (2) with an addition of the AUS subgroup of category III to those indications. The associations between UP-HT and BSRTC categories indicating the necessity of surgical treatment were evaluated with the use of the logistic regression analysis. Odds ratios (OR) with relative 95% confidence intervals (95% CI) were calculated to determine the relevance of potential predictors of the outcome. The comparison of frequency distributions was performed with the chi2 test (with modifications appropriate for the number of analyzed cases). The Kruskal-Wallis test was used for comparing continuous variables between groups. The value of 0.05 was assumed as the level of significance. The statistical analysis was performed with the Dell Statistica (data analysis software system), version 13, Dell Inc. (2016), Round Rock, TX, U.S.

The study design was approved by the Local Bioethics Committee (approval code RNN/151/17/KE) and all the patients gave their informed consent.

## 3. Results

Table 1 shows the distribution of particular variants of UP-HT among the examined patients. The most frequent variant was “marked hypoechoic, heterogeneous echotexture” (28.0%), followed by “normoechoic pseudo-nodular areas” (21.4%), and “multiple, discrete marked hypoechoic areas” (18.5%). The least frequent variant was “echostructure similar to connective tissue” (0.9%). Patients with “hypoechoic, heterogeneous echotexture” and “multiple, discrete marked hypoechoic areas” variants were significantly younger than patients with the “normoechoic pseudo-nodular areas” variant. The oldest group of patients was characterized by the “echostructure similar to connective tissue” variant. All the patients with the latter variant were treated with levothyroxine due to hypothyroidism. Apart from that variant, the levothyroxine treatment was most frequent in patients with “marked hypoechoic, heterogeneous echotexture” (76.3%) and least frequent in the case of the thyroid gland with no signs of HT (17.4%). The percentage of males was highest in patients with the “heterogeneous echotexture with hyperechoic, fibrous septa” variant (12.5%), while in other subgroups it was under 6% (NS). The thyroid volume was the lowest in the case of the “echostructure similar to connective tissue” variant, and the highest in patients with the “normoechoic pseudo-nodular areas” variant. We did not find any significant differences in the mean volume of biopsied nodules, but cytologically examined nodules under 1 cm in diameter were least frequent in the “normoechoic pseudo-nodular areas” variant (5.2%) (Appendix A). That variant was also more frequently associated with a focal HT than other variants (Table 2). The levothyroxine treatment was less frequently applied in patients with focal HT than those with a diffuse disease (66.7% vs. 51.5%, *p* < 0.05). Compared with other UP-HT, the “multiple, discrete marked hypoechoic areas” variant was more often associated with hypervascularity of the parenchyma (19.4%), and less often with typical lymph nodes (28.2%). Hypovascularity of the parenchyma was most commonly observed in the “echostructure similar to connective tissue” variant (60.0%), and it was also more frequent in patients with the “heterogeneous echotexture with hyperechoic, fibrous septa” variant (18.8%) compared to other UP-HT. Additional signs of HT (such as typical lymph nodes or changed vascularity) were most commonly absent in the “hypoechoic, homogenous echostructure” variant (43.9%), and least commonly in the “multiple, discrete marked hypoechoic areas” variant (21.4%), and the “echostructure similar to connective tissue” variant (one out of five cases). 

FNA outcomes were classified into one of the categories of diagnostic smears (categories II-VI of BSRTC) in 87.1% of cases. Non-diagnostic outcomes (category I of BSRTC: 12.9%) were most frequent in patients with the “echostructure similar to connective tissue” variant (80.0%), significantly more frequent than in all the other variants. Microscopic features of HT were present in 71.8% (348 out of 485) of smears with the diagnostic material, the least often in the “hypoechoic, homogenous echostructure” variant (45.2%, *p* < 0.05 vs. other variants, except for the “thyroid with no signs of HT” variant—60.0%). In other variants, the incidence of microscopic HT features ranged from 67.1% to 83.3% (not taking into account the presence of these features in the single diagnostic cytology obtained in the “echostructure similar to connective tissue” variant). Patients with microscopic features of HT were treated with levothyroxine more often than patients without those features: 67.5% (235 out of 348) vs. 52.5% (72 out of 137), *p* < 0.05.

Cytological outcomes of categories IV to VI constituted 1.6% of all the results (Table 3). In the case of category IV, a suspicion of Hurthle cell tumor was formulated in all patients, and in the case of categories V and VI, PTC was suspected or diagnosed respectively. Category III constituted 15.6% of all results, and 90.8% of smears of that category showed features of architectural atypia (FLUS), while the remaining 9.2%—features of nuclear atypia (AUS). The diagnoses of a benign lesion (BSRTC category II) were most frequent in “thyroid with no signs of HT” and “normoechoic pseudo-nodular areas” variants (>78%).

Indications for a surgical treatment resulting from the FNA outcome (categories IV–VI) were identified only in patients with one of three UP-HT: “Multiple, discrete marked hypoechoic areas”, “hypoechoic, heterogeneous echotexture” or “marked hypoechoic, heterogeneous echotexture”. When the diagnosis of AUS was also considered as an indication for surgery (option 2), one such case was found in the “homogeneous, hypoechoic echotexture” variant. In total, taking the wider option 2 into consideration, indications for surgery based on the FNA outcome were found in 17 patients (3.1% of all examined with FNA and 3.5% of patients with a diagnostic FNA outcome). So far, 13 of them have been operated, including all patients with FNA outcomes of BSRTC category VI and V (PTC was confirmed in all cases), one out of two patients with the outcome of category IV (a benign lesion was found) and five patients with AUS (cancers were found in two of them: PTC and medullary thyroid carcinoma—MTC). The percentage of nodules with the diameter under 1 cm was nearly three times higher among nodules with cytological indications for surgical treatment (categories IV–VI and subcategory AUS of category III) than in the whole examined group—29.4% (five out of 17) vs. 10.1% (*p* < 0.01) (see Appendix A).

The logistic analysis of regression confirmed that the presence of the “multiple, discrete marked hypoechoic areas” variant significantly increased the odds of obtaining a cytological outcome which would be an indication for surgical treatment—regardless of the mode of analysis (option 1, OR: 5.7; option 2, OR: 4.2) (Table 4). In the case of option 2, the “hypoechoic, heterogeneous echotexture” variant was also found to increase the odds (OR: 2.7) nearly significantly. On the other hand, the presence of “normoechoic pseudo-nodular areas” variant significantly increased the odds for the cytological diagnosis of a benign lesion, OR (95% CI): 1.7 (1.1–2.8), *p* = 0.027. No significant effect of age or gender on the chance of obtaining an FNA outcome indicating the need for surgical treatment was found.

There were UP-HT variants in which FNA outcomes indicating the necessity of surgery were never observed. If FNA had not been performed in patients with such variants, then the number of performed FNA would have been limited to 378 (67.9%).

## 4. Discussion

Hashimoto′s thyroiditis poses significant problems for the morphological diagnostics of the thyroid. Heterogeneity of the parenchyma affected by HT hampers the identification of nodules during US examination and the marked percentage of equivocal FNA outcomes often compels repetition of the biopsy [12,13,14,15]. For these reasons, we decided to find out whether the US pattern of thyroid parenchyma affected by HT brings any significant information on the risk of malignancy in co-existing nodules. According to our knowledge, so far there has been no attempt to relate the appearance of thyroid parenchyma in patients with HT to the frequency of cytological reports of a suspicious or malignant category. There are some reports on features of the ultrasound image which are relevant for confirming the diagnosis of HT or arousing suspicion of an incidental diffuse thyroid disease, as well as diagnosing thyroid lymphoma [7,18,19,27,28,29].

Our study has shown that there are significant differences in the frequency of obtaining suspicious and malignant cytological diagnoses in relation to the UP-HT variant. These differences are distinct despite the fact that in our population, which had been exposed to iodine deficiency for a long time, non-neoplastic nodules and consequently the diagnoses of benign lesions are predominant [26]. Cytological reports of categories IV-VI constituted only 1.6% of all the outcomes and the majority of suspicious or malignant FNA results were related to PTC, which is concordant with the literature [13,16]. However, cytological outcomes of category III constituted as much as 15.6% of all FNA and 17.9% of diagnostic FNA, over twice as much as in the general population examined at our center—with or without HT (6.4% of diagnostic FNA) [26]. Smears with features of architectural atypia (FLUS) constituted just over 90% of the cases in that category, while smears with features of nuclear atypia characteristic for PTC (AUS) were the remaining 9.2%. That percentage was markedly higher than in the general population, in which the percentage of AUS within category III was 2% [26]. This is a result of the above-mentioned predomination of non-neoplastic lesions in our patients that developed as a consequence of iodine deficiency. Many published reports as well as our observations indicate that the risk of malignancy in the case of AUS subcategory is at least two-fold higher than in the case of the FLUS subcategory [26,30]. For this reason, we regarded the AUS diagnosis as an indication for a surgical treatment in one option of the analysis. In addition, as if to confirm this approach, a cancer was revealed in two out of five (40.0%) surgically treated patients with that diagnosis. We found FNA results of categories IV–VI, which are usually regarded as an obvious indication for a surgical treatment, only in three UP-HT variants: “Multiple, discrete marked hypoechoic areas”, “hypoechoic, heterogeneous echotexture”, and “marked hypoechoic, heterogeneous echotexture”. Additionally, only the “multiple, discrete marked hypoechoic areas” variant increased the risk of obtaining a cytological outcome that indicated the surgical treatment was more than four-fold. It is noteworthy that the same variant was the most frequently accompanied by hypervascularity of the parenchyma and less often by characteristic lymph nodes (oval or round shape, hypoechoic with an absence of hilum but without malignancy suggested features). Lymph nodes of that type were found significantly more often in the “normoechoic pseudo-nodular areas” variant, which increased the chance of obtaining the diagnosis of a benign lesion nearly twice. Other authors did not correlate the incidence of reactive lymph nodules (variably defined) with UP-HT variants [7,31].

Several reports indicated that the image of the thyroid described as micronodules ranging in size from 1 to 7 mm, and so corresponding to our “multiple, discrete marked hypoechoic areas” variant, was highly diagnostic of HT and connected to its early phase [6,7,32]. Our observations showed that the frequency of the treatment for hypothyroidism was lower in that variant than in “marked hypoechoic, heterogeneous echotexture” and “echostructure similar to connective tissue” variants, which might support that suggestion. The evolution of the image of the parenchyma in the course of HT is an additional hindrance in the diagnostics of patients with HT. We did not analyze the relation between the UP-HT variant and the time from the onset of the disease directly due to the lack of reliable data on the onset of HT. It is a common situation in clinical practice since the onset of hypothyroidism can be subtle and difficult to locate in time precisely. However, several of our observations (such as higher mean age of patients, more frequent treatment of hypothyroidism, small thyroid volume, and its diminished vascularity) strongly suggest that the “echostructure similar to connective tissue” variant is typical of advanced stages of the disease, as is the “marked hypoechoic, heterogeneous echotexture” variant. On the contrary, “hypoechoic, heterogeneous echotexture” and “multiple discrete marked hypoechoic areas” variants are characteristic of its early phase. The “normoechoic pseudo-nodular areas” variant showed an apparent discrepancy between the high mean age of patients and the low percentage of those treated with levothyroxine. It may be related to the fact that the above-mentioned variant was identified in the focal type of HT more often than other variants. Accordingly, Anderson et al. claimed that the focal HT is a clinically less severe form of the disease than the diffuse HT [6].

On the whole, the “multiple discrete marked hypoechoic areas” variant, which is related to the higher risk of an alarming FNA outcome and is typical of the early phase of the disease, should prompt a more vigilant follow-up of patients. Similar clinical implications could be assigned to the “hypoechoic, heterogeneous echotexture” variant which is also characteristic of the early phase. Contrarily, the presence of other variants, especially the “normoechoic pseudo-nodular areas” one, should prompt a less rigorous attitude to FNA performing, especially when the US risk of malignancy in a nodule is low. It should be stressed that the presence of US malignancy risk features is of primary importance for the identification of indications for the biopsy. Our patients were subjected to FNA according to the current guidelines based on the analysis of those features. Nodules with a maximal diameter under 1 cm were biopsied only in the case of a worrisome US image. Consequently, the percentage of such nodules was three times higher in patients with an FNA outcome that was an indication for a surgical treatment than in other patients. It cannot be left out that some of the biopsied lesions in the “normoechoic pseudo-nodular areas” variant were in fact large pseudonodules (the number of biopsied lesions with the maximal diameter <1 cm was the lowest in that UP-HT variant). The analysis of US malignancy risk features of nodules did not fit the scope of the present study. However, published reports usually confirm their usefulness in patients with HT in a standard version or with slight modifications [8,9,10,33,34,35,36].

Our study has several limitations. It included patients treated in an out-patient clinic with the clinical diagnosis of HT. The data on the intensity of symptoms and laboratory test results were not standardized and thus hardly comparable. We could not consider the ultrasound image of the thyroid or the cytological outcome as criteria for the diagnosis of HT since precisely those features were analyzed in our study. For that reason, we decided to assume the elevated concentration of TPOab as the criterion confirming the diagnosis of HT. Elevated TPOab are regarded as sensitive and an especially specific diagnostic tool in the case of HT. However, we are aware that an abnormal TPOab concentration may be found in healthy individuals, as well as in patients with Graves′ disease, and that US features of HT may precede TPOab positivity [2,3,37]. Our data show that the microscopic examination, which is regarded as a gold standard by some investigators, is not fully reliable either. Features of HT in smears may not be present in some patients with other features of HT such as hypothyroidism or abnormal TPOab. The advantage of our study is its prospective design. Investigators who classified ultrasound images into UP-HT types did not know the FNA outcome of the patient. We recognize that ultrasound imaging is an operator-dependent examination with subjective interpretation and that the recognition of particular US patterns may vary by the ultrasonographer. That is the reason an assessment of US patterns in HT should be unified in the diagnostic center.

## 5. Conclusions

In conclusion, the US examination of the thyroid should include the precise evaluation of the parenchyma. It not only helps suggest or confirm the diagnosis of HT but it may also optimize the diagnostic process. The identification of the presence of “multiple, discrete marked hypoechoic areas” in thyroid parenchyma is an additional risk factor of malignancy in a co-existing thyroid nodule. Furthermore, the presence of “normoechoic pseudo-nodular areas” speaks in favor of the benignity of co-existing nodules. This issue should be further investigated with a detailed analysis of US malignancy risk features of nodules in relation to the ultrasound pattern of the surrounding parenchyma.

## Figures and Tables

**Table 1 jcm-10-00638-t001:** Distribution of variants of ultrasonographic patterns of thyroid parenchyma in patients with Hashimoto′s thyroiditis (UP-HT) subjected to the fine needle aspiration (FNA) of thyroid nodules, including demographic data, information on the treatment with levothyroxine (LT4), and thyroid volume.

UP-HT Variant	No./% of Patients—*p*	Mean Age ± SD (Years)—*p*	No./% of LT4 Treated Patients—*p*	Mean Thyroid Volume ± SD (cm^3^)—*p*
(a) hypoechoic, homogeneous echotexture	41/7.4 *<0.0001* vs. *c,e,f,g;* *<0.001* vs. *b; <0.05* vs. *h*	59.9 ± 15.5 *NS*	24/58.5 *<0.005* vs. *h; <0.05* vs. *c*	16.0 ± 10.3 *<0.0001* vs. *f*
(b) hypoechoic, heterogeneous echotexture	78/14.0 *<0.0001* vs. *c,d,g,h; <0.001* vs. *a;* *<0.05* vs. *e,f*	53.4 ± 15.1 *<0.005* vs. *f*	51/65.4 *<0.001* vs. *h*	16.6 ± 8.9 *<0.0001* vs. *f*
(c) marked hypoechoic, heterogeneous echotexture	156/28.0 *<0.0001* vs. *a,b,d,g,h* *<0.001* vs. *e; <0.05* vs. *f*	58.8 ± 14.6 *<0.005* vs. *e*	119/76.3 *<0.0001* vs. *h; <0.01* vs. *e;* *<0.05* vs. *a,f*	13.8 ± 7.0 *<0.0001* vs. *f;* *<0.005* vs. *d,e*
(d) heterogeneous echotexture, hyperechoic, fibrous septa	32/5.7 *<0.0001* vs. *b,c,e,f,g*	56.4 ± 15.5 *NS*	21/65.6 *<0.005* vs. *h*	20.8 ± 12.0 *<0.005* vs. *c*
(e) multiple, discrete marked hypoechoic areas	103/18.5 *<0.0001* vs. *a,d,g,h* *<0.001* vs. *c; <0.05* vs. *b*	52.2 ± 13.3 *<0.0001* vs. *f;* *<0.005* vs. *c*	63/61.2 *<0.001* vs. *h; <0.01* vs. *c*	17.8 ± 8.3 *<0.005* vs. *c;* *<0.01* vs. *f*
(f) normoechoic pseudo-nodular areas	119/21.4 *<0.0001* vs. *a,d,g,h; <0.05* vs. *b,c*	62.3 ± 11.0 *<0.0001* vs. *e;* *<0.005* vs. *b*	74/62.2 *<0.001* vs. *h; <0.05* vs. *c*	25.3 ± 16.0 *<0.0001* vs. *a,b,c; <0.01* vs. *e,g*
(g) echostructure similar to connective tissue	5/0.9 *<0.0001* vs. *a,b,c,d,e,f; <0.001* vs. *h*	67.6 ± 11.1 *NS*	5/100.0 *<0.005* vs. *h*	7.0 ± 5.7 *<0.01 vs. f*
(h) thyroid with no signs of HT	23/4.1 *<0.0001* vs. *b,c,e,f;* *<0.001* vs. *g; <0.05* vs. *a*	52.6 ± 14.3 *NS*	4/17.4 *<0.0001* vs. *c; <0.001* vs. *b,e,f; <0.005* vs*. a,d,g*	19.8 ± 10.3 *NS*
Total	557/100.0	57.4 ± 14.1	361/64.8	18.2 ± 11.6

**Table 2 jcm-10-00638-t002:** Detailed characteristics of ultrasound image of the thyroid parenchyma in patients with Hashimoto′s thyroiditis (UP-HT) with regard to the vascularity and presence of lymph nodes typical of HT.

UP-HT Variant	Diffuse HT (No./%) *p*	Typical Lymph Nodes (No./%) *p*	Hypovascularity (No./%) *p*	Hypervascularity (No./%) *p*	No Typical Lymph Nodes Nor Abnormal Vascularity (No./%) *p*
(a) hypoechoic, homogeneous echotexture	41/100.0 *<0.0001* vs. *f; <0.05* vs*. e*	19/46.3 *<0.05* vs. *e*	1/2.4 *<0.0005* vs. *g*	2/4.9 *NS*	18/43.9 *<0.01* vs. *e; <0.05* vs. *c*
(b) hypoechoic, heterogeneous echotexture	77/98.7 *<0.0001* vs. *f; <0.05* vs. *e*	30/38.5 *NS*	3/3.9 *<0.0001* vs. *g; <0.05* vs. *d*	9/11.5 *NS*	26/33.3 *<0.05* vs. *h*
(c) marked hypoechoic, heterogeneous echotexture	148/94.9 *<0.0001* vs. *f; <0.05* vs. *e*	56/36.9 *NS*	10/6.4 *<0.0005* vs. *g; <0.05* vs. *d*	16/10.3 *<0.05* vs. *e*	42/26.9 *<0.01* vs. *h; <0.05* vs. *a*
(d) heterogeneous echotexture, hyperechoic, fibrous septa	29/90.6 *<0.05* vs. *f*	14/43.8 *NS*	6/18.8 *<0.01* vs. *f; <0.05* vs. *b,c,e*	4/12.5 *NS*	8/25.0 *<0.01* vs. *h*
(e) multiple, discrete marked hypoechoic areas	90/87.4 *<0.0001* vs. *f; <0.05* vs. *a,b,c*	29/28.2 *<0.01* vs. *f;<0.05* vs. *a*	5/4.9 *<0.0005* vs. *g; <0.05* vs. *d*	20/19.4 *<0.05* vs. *c,h*	22/21.4 *<0.0001* vs. *h;* *<0.01* vs. *a,f*
(f) normoechoic pseudo-nodular areas	76/63.9 *<0.0001* vs. *a,b,c,e; <0.05* vs. *d,h*	55/46.2 *<0.01* vs. *e*	5/4.2 *<0.01* vs. *d*	17/14.3 *NS*	45/37.8 *<0.01* vs. *e; <0.05* vs. *h*
(g) echostructure similar to connective tissue	5/100.0 *NS*	3/60.0 *NS*	3/60.0 *<0.0001* vs. *b;* *<0.0005* vs. *a,c,e; <0.005* vs. *h*	0/0.0 *NS*	1/20.0 *NS*
(h) thyroid with no signs of HT	23/100.0 *<0.05* vs. *f*	14/60.9 *NS*	0/0.0 *<0.005* vs. *g*	0/0.0 *<0.05* vs. *e*	14/60.9 *<0.0001* vs. *e; <0.01* vs*. c,d; <0.05* vs. *b,f*
Total	489/87.8	220/39.5	33/5.9	68/12.2	176/31.6

**Table 3 jcm-10-00638-t003:** Distribution of the categories of FNA outcomes of the thyroid nodules according to the Bethesda System for Reporting Thyroid Cytology (BSRTC) in 557 patients with Hashimoto′s thyroiditis in relation to the ultrasound pattern of the thyroid parenchyma (UP-HT).

UP-HT Variant	Category of BSRTC (No./%) *p*							
	I	II	III	*FLUS* */AUS*	IV	V	VI	Total
(a) hypoechoic, homogeneous echotexture	10/24.4 *<0.01* vs. *g*	24/58.5	7/17.1	*6/1*	-	-	-	41
(b) hypoechoic, heterogeneous echotexture	8/10.3 *<0.0001* vs. *g* *<0.05* vs. *a*	52/66.7	15/19.2	*13/2*	-	1/1.3	2/2.6	78
(c) marked hypoechoic, heterogeneous echotexture	25/16.0 *<0.0001* vs. *g*	107/68.6	23/14.7	*21/2*	1/0.6	-	-	156
(d) heterogeneous echotexture, hyperechoic, fibrous septa	2/6.3 *<0.0001* vs. *g*	21/65.6	9/28.1	*9/-*	-	-	-	32
(e) multiple, discrete marked hypoechoic areas	10/9.7 *<0.0001* vs. *g* *<0.05* vs. *a*	73/70.9	15/14.6	*12/3*	1/1.0	3/2.9	1/1.0	103
(f) normoechoic pseudo-nodular areas	10/8.4 *<0.0001* vs. *g*	93/78.2 *<0.05* vs. *a,g*	16/13.4	*16/-*	-	-	-	119
(g) echostructure similar to connective tissue	4/80.0	1/20.0	-	*-*	-	-	-	5
(h) thyroid with no signs of HT	3/13.0 *<0.001* vs. *g*	18/78.3	2/8.7	*2/-*	-	-	-	23
Total	72/12.9	389/69.8	87/15.6	*79/8*	2/0.4	4/0.7	3/0.5	557

**Table 4 jcm-10-00638-t004:** Indications for the surgical treatment on the basis of FNA outcome in relation to the variant of the ultrasound pattern of the thyroid parenchyma (UP-HT) —results of the logistic analysis of regression (OR—odds ratio of the necessity for the surgical treatment).

UP-HT Variant	Indications for the Surgical Treatment Based on the BSRTC Category							
	1st Option				2nd Option			
	Absent: II-III (No./%)	Present: IV-VI (No./%)	*p*	OR (95% CI) *p*	Absent: II+FLUS (No./%)	Present: AUS+IV-VI (No./%)	*p*	OR (95% CI) *p*
(a) hypoechoic, homogeneous echotexture	31/100.0	-		-	30/96.8	1/3.2		0.8 (0.1–1.6) *0.813*
(b) hypoechoic, heterogeneous echotexture	67/95.7	3/4.3		3.1 (0.8–12.9) *0.110*	65/92.9	5/7.1	*<0.05* vs. *f*	2.7 (0.9–7.8) *0.073*
(c) marked hypoechoic, heterogeneous echotexture	130/99.2	1/0.8		0.3 (0.1–2.6) *0.281*	128/97.7	3/2.3		0.5 (0.1–1.9) *0.341*
(d) heterogeneous echotexture, hyperechoic, fibrous septa	30/100.0	-		-	30/100.0	-		
(e) multiple, discrete marked hypoechoic areas	88/94.6	5/5.4	*<0.05* vs. *f*	5.7 (1.5–21.8) *0.010*	85/91.4	8/8.6	*<0.01* vs. *f*	4.2 (1.6–11.1) *0.004*
(f) normoechoic pseudo-nodular areas	109/100.0	-	*<0.05* vs. *e*	-	109/100.0	-	*<0.01* vs. *e* *<0.05* vs. *b*	-
(g) echostructure similar to connective tissue	1/100.0	-		-	1/100.0	-		-
(h) thyroid with no signs of HT	20/100.0			-	20/100.0	-		-

## Data Availability

The data presented in this study are available on request from the corresponding author. The data are not publicly available due to patient privacy restrictions.

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
