# Peer review of "The Presence of Hypoechoic Micronodules in Patients with Hashimoto′s Thyroiditis Increases the Risk of an Alarming Cytological Outcome"

_jcm, 2021, doi:10.3390/jcm10040638_

Round 1

Reviewer 1 Report

I congratulate you on this study. Well designed and presented well despite the obvious limitations and biases of selection.

1)Would be interested to know the progression of these nodules in the setting of HT as active surveillance is being considered to nodule size upto 1.5 cm in PTC. If there is any data on that could you please publish that as a supplement.

2)Is there any data on LN biopsy and involvement in the setting of HT?

Author Response

We sincerely thank for complimenting our study.

  1. Would be interested to know the progression of these nodules in the setting of HT as active surveillance is being considered to nodule size up to 1.5 cm in PTC. If there is any data on that could you please publish that as a supplement.

Indeed, the question of active surveillance of small PTC is widely discussed. In the case of patients with HT this discussion is particularly justified as there are reports suggesting that PTC in these patients shows a more benign course and better prognoses. However, the recommendations in our country do not allow for conservative treatment in the case of an FNA outcome that indicates PTC. Consequently, all patients with such a diagnosis are referred to surgery. In the few cases of patients who do not agree to surgical treatment or such treatment is contraindicated for other reasons, we have not observed any significant enlargement of the nodule. There is one particularly interesting patient – a currently 80-year-old female with the diagnosis of PTC who has been observed for 9 years with no progression noted. Obviously, we cannot be sure if the lesion is really PTC and not a kind of borderline tumor, for example, as the diagnosis could not be confirmed histopathologically.

  1. Is there any data on LN biopsy and involvement in the setting of HT?

We have no experience of our own in the use of LN biopsy. Published data indicate that LN biopsy has a high specificity for the diagnosis of malignancy in the thyroid nodules with initially indeterminate results on FNA. But according to our knowledge, those studies did not analyze patients with HT separately. There are reports suggesting that LN biopsy has low non-diagnostic result rates. This provides the rationale for performing an evaluation of the usefulness of LN biopsy in patients with HT, as thyroid fibrosis is quite often a cause of the non-diagnostic outcome of FNA in the course of chronic thyroiditis.

Reviewer 2 Report

Methods paragraph should be improved. There are some discrepancies that affect the value of presented findings. 

In particular, it has been stated that the biopsy was performed on thyroid nodules with a diameter of at least 5 mm. However, group 5 includes  multiple, discrete marked hypoechoic areas with micronodules - sized as 1 to 6 mm. Please explain. 

The mean size of analysed nodules according to categories has not been reported

The diagnostic categories of the Bethesda system should be fully described given the key role in the statistical analysis. 

Please use always the same code to indicate groups. If I understood in the text Authors use arabic numbers and tin the tables they use letters.

In the results, it has been stated that FNA outcomes were classified into one of diagnostic categories in 87.1% of cases. Please explain better for each category. What about the remaining 12.9%?

"If FNA had been performed only in patients with UP-HT variants in which FNA outcomesindicating the surgery were observed then the number of performed FNA could have been limited (67.9%)." This sentence is not clear. Please rephrase.

Author Response

  1. Methods paragraph should be improved. There are some discrepancies that affect the value of presented findings. In particular, it has been stated that the biopsy was performed on thyroid nodules with a diameter of at least 5 mm. However, group 5 includes  multiple, discrete marked hypoechoic areas with micronodules - sized as 1 to 6 mm. Please explain. 

We did not biopsy the discrete marked hypoechoic areas that were characteristic of the fifth (e.) variant of UP-HT, despite the fact that their size (but usually not all three diameters) might have in some cases exceeded the lower size limit of nodules subjected to FNA by 1 mm. Marked hypoechoic areas that are typical of this variant occur in the thyroid in high numbers and do not present ultrasound malignancy risk features (such as microcalcifications, irregular margins or suspicious shape), so they can be distinguished from very small suspicious nodules. Additionally, at least one of their diameters is usually under 5 mm. To clarify this point we have added the following sentence: “Nodules < 1 cm were biopsied only if they showed particularly worrying ultrasound image and none of their diameters were under 5 mm” just after the sentence “The biopsy was performed on thyroid nodules with a diameter of at least 5 mm (and usually over 1 cm) and at least one malignancy risk factor (US or clinical), according to the recommendations in 4effect in our country [24-25]” in the Material and methods section.

  1. The mean size of analysed nodules according to categories has not been reported

It is unclear to us whether this remark refers to the mean size of analysed nodules according to the type of UP-HT or according to the categories of the Bethesda system. If the former is true, there is a table in the supplementary material which shows the mean volume of biopsied nodules for the evaluated variants of UP-HT and these data are also mentioned in the main part of the paper (lines 152-155).

In the revised version we added another table to the supplementary material that shows data on mean sizes of nodules of each category of the Bethesda system. It also contains the exact number of biopsied nodules under 1 cm in size in each BSRTC category. The table has been referred to in the Results section at the end of the sentence in lines 214-216. The sentence has been also extended to mark clearly that it refers to the second option of the analysis of indications to surgical treatment:

The percentage of nodules with the diameter under 1 cm was nearly three times higher among nodules with cytological indications for surgical treatment (categories IV-VI and subcategory AUS of category III) than in the whole examined group - 29.4% (five out of 17) vs. 10.1% (p<0.01) (Table S2).”

  1. The diagnostic categories of the Bethesda system should be fully described given the key role in the statistical analysis. 

In Material and methods section the paragraph describing that point (lines 105-110) has been significantly extended from:

 “Smears were fixed with 95% ethanol solution and stained with haematoxylin and eosin. A detailed description of the classification of nodules into specific diagnostic categories of the Bethesda system was presented in our earlier report [26]. When one patient underwent FNA of two or more nodules, the nodule diagnosed with the highest Bethesda System for Reporting Thyroid Cytology (BSRTC) category was considered for the analysis of relation between FNA outcome and UP-HT.”

to:

Smears were fixed with 95% ethanol solution and stained with haematoxylin and eosin. The FNA outcome of each nodules was classified into one of six categories in the Bethesda System for Reporting Thyroid Cytology (BSRTC). In this classification, category I includes non-diagnostic/unsatisfactory biopsies, category II: benign lesions, category III: follicular lesion of undetermined significance (FLUS) or atypia of undetermined significance (AUS), category IV: suspicious for a follicular neoplasm, category V: suspicious for malignancy and category VI: malignant neoplasm. Categories IV, V and VI are regarded as an indication for surgical treatment because of the high risk of malignancy. Category III presents the most diverse risk of malignancy. At our center, similarly to majority of others, that risk is significantly higher for diagnoses of AUS than FLUS. A detailed description of the classification of nodules into specific diagnostic categories of the Bethesda system as well as the risk of malignancy related to particular categories at our center were presented in our earlier report [26]. When one patient underwent FNA of two or more nodules, the nodule diagnosed with the highest BSRTC category was considered for the analysis of relation between FNA outcome and UP-HT.”

  1. Please use always the same code to indicate groups. If I understood in the text Authors use arabic numbers and tin the tables they use letters.

Thank you for pointing out this inconsequence. Following the suggestion we have replaced Arabic numbers with letters to enumerate evaluated ultrasound patterns of HT (lines 82-87).

  1. In the results, it has been stated that FNA outcomes were classified into one of diagnostic categories in 87.1% of cases. Please explain better for each category. What about the remaining 12.9%?

The term “diagnostic category” that we used was imprecise and potentially misleading. So the sentences in question (lines 174-176) have been rewritten as follows:

FNA outcomes were classified into one of the categories of diagnostic smears (categories II-VI of BSRTC) in 87.1% of cases. Non-diagnostic outcomes (category I of BSRTC: 12.9%) were most frequent in patients with “echostructure similar to connective tissue” variant (80.0%), significantly more frequent than in all other variants.

  1. "If FNA had been performed only in patients with UP-HT variants in which FNA outcomes indicating the surgery were observed then the number of performed FNA could have been limited (67.9%)." This sentence is not clear. Please rephrase.

The sentence has been rephrased as follows:
There were UP-HT variants in which FNA outcomes indicating the necessity of surgery were never observed. If FNA had not been performed in patients with such variants, then the number of performed FNA would have been limited to 378 (67.9%).

Round 2

Reviewer 2 Report

Authors have addressed the major issues highlighted in the previous review